# Control of Acute Arboviral Infection by Natural Killer Cells

**DOI:** 10.3390/v11020131

**Published:** 2019-01-31

**Authors:** Christopher Maucourant, Caroline Petitdemange, Hans Yssel, Vincent Vieillard

**Affiliations:** 1Sorbonne Université, UPMC Univ Paris 06, Inserm U1135, CNRS ERL8255, Centre d’Immunologie et des Maladies Infectieuses (CIMI-Paris), 75013 Paris, France; maucourantchristopher@hotmail.fr (C.M.); hans.yssel@inserm.fr (H.Y.); 2Institut Gustave Roussy, CNRS UMR9196, Unité Physiologie et Pathologie Moléculaires des Rétrovirus Endogènes et Infectieux, 94800 Villejuif, France; caroline.petitdemange@gmail.com

**Keywords:** natural killer cells, arboviruses, interferon-γ, cytotoxicity

## Abstract

The recent explosive pandemic of chikungunya virus (CHIKV) followed by Zika (ZIKV) virus infections occurring throughout many countries represents the most unexpected arrival of arthropod-borne viral diseases in the past 20 years. Transmitted through the bite of *Aedes* mosquitoes, the clinical picture associated with these acute arbovirus infections, including Dengue (DENV), CHIKV and ZIKV, ranges from classical febrile illness to life-threatening disease. Whereas ZIKV and CHIKV-mediated infections have previously been recognized as relatively benign diseases, in contrast to Dengue fever, recent epidemic events have brought waves of increased morbidity and mortality leading to a serious public health problem. Although the host immune response plays a crucial role in controlling infections, it may also promote viral spread and immunopathology. Here, we review recent developments in our understanding of the immune response, with an emphasis on the early antiviral immune response mediated by natural killer cells and emphasize their Janus-faced effects in the control of arbovirus infection and pathogenesis. Improving our understanding knowledge on of the mechanisms that control viral infection is crucial in the current race against the globalization of arbovirus epidemics.

## 1. Introduction

### 1.1. Emerging Arboviruses: A Global Public Health Threat

Pathogens transmitted by mosquitoes constitute a profound and growing health burden according to the World Health Organization (http://www.who.int/mediacentre/factsheets/fs387/en/). Urbanization, poverty, globalization, migration and climate changes are widely recognized as contributors to the spread of infections by arthropod-borne viruses (arboviruses). These latter are genetically highly diverse and represent one of the largest virus groups, with nearly 500 members known at present throughout the world. While many arboviruses do not currently appear to be pathogens, they provide an immense resource for the emergence of new pathogens in the future To date, the most medically important mosquito-borne arboviruses are grouped in two major families: (i) the *Flaviviridae* (genus *flavivirus*) which includes West Nile (WNV), yellow fever (YFV), dengue (DENV) and Zika (ZIKV) viruses and (ii) the *Togaviridae* (genus *alphavirus*) to which belong Chikungunya (CHIKV), Ross River (RRV) and Sindbis (SINV) viruses [1,2]. A more complete list of arboviruses and their geographical repartition is shown in Figure 1. Of concern, the frequency and magnitude of arboviral epidemics has increased in both established and recently colonized geographic areas. Globally, millions of people are infected each year by arboviruses that cause epidemics, spread by the day-biting of the *Aedes (Ae.) aegypti* mosquito, which is the primary vector [3]. This mosquito evolved from the sylvan African *Ae. formosus* to become an anthropophilic species that breeds in urban environments and feeds primarily on humans [4]. In contrast to the typically sylvatic outbreaks of CHIKV that occur in Africa, a single amino acid mutation in the E1 envelop protein adapted the CHIKV to *Ae. albopictus*, a vector that readily adapts to diverse environmental conditions. Importantly, the past three decades have seen a dramatic global expansion in the geographic distribution of *Ae. albopictus* that continues today [5,6].

The intensification of the globalization process has resulted in a sharp increase in the spread of these infectious diseases with a staggering economic burden. For example, DENV causes more than 50 million infections yearly with more than 13,000 fatal cases for an annual global cost of US $ 9 billion [7]. In addition, the recent outbreaks of ZIKV, associated with neurological disorders and neonatal malformations in Latin America, YFV outbreaks in Angola and Brazil, WNV in North America, as well as the emergence of CHIKV from sub-Saharan Africa in the not-too-distant past and its relatively recent arrival in the Americas and Europe have propelled arboviruses in the news and placed them at the top of social, political and public health agendas. The intensification of the globalization process has resulted in a sharp increase in the spread of infectious diseases to populations lacking native immunity.

### 1.2. Host Immune Responses to Mosquito Bites and Arbovirus Infection

Despite their considerable diversity, mosquito-borne viruses share a common attribute: transmission via the skin at the site of the mosquito bite. Figure 2 shows that after the bite, the majority of the virus is directly deposited into the extracellular space of the dermis, which represents the first stage of infection. Both DENV and ZIKV have been shown to infect dermal dendritic cells (DCs) and although there are no reports of YFV infecting Langerhans cells, it can nevertheless infect myeloid DCs. Viral entry into susceptible cells during ZIKV infection is mediated by DC-SIGN but appears to be DC-SIGN-independent in the case of DENV and YFV [8]. It has been shown that CHIKV is able to replicate in epithelial and endothelial cells and, to a lesser extent, monocyte-derived macrophages and that viral entry into these cells was mediated by several receptors including prohibitin, phosphatidylserine-mediated virus entry-enhancing receptors and glycosaminoglycans [9]. Although the host rapidly mounts a response to control the virus in the dermis, the virus is able to disseminate quickly to different relevant lymphoid and non-lymphoid tissues via the peripheral blood (Figure 2). In a zebrafish model, it was shown that CHIKV rapidly disseminates to various organs within approximately 14 h after infection [10]. During this silent incubation period, the viral load in the circulation increases rapidly to reach a high serum levels of infectious particles. The acute phase of arbovirus infection is accompanied by an early type I interferon (IFN) response [11]. Indeed, mice defective in IFNAR signalling succumb to most arbovirus infections within a few days [12].

Concomitantly with the production of type I IFN, trauma associated with mosquito bites and the presence of salivary protein(s) induce a multistep recruitment of leukocytes that is initiated with mast cell degranulation and the recruitment of neutrophils that express high levels of the key proinflammatory cytokine interleukin-1β, which are important for the induction of inflammatory responses [13,14]. It is important, however, to note that results obtained from cytokine analyses in patients with arbovirus infection are often contradictory. Thus, For the same viral infection, discrepancies between studies could reflect different genetic backgrounds, heterogeneous sanitary conditions, patient management strategies or differences in laboratory techniques [6,15]. The inflammatory process that occurs following the spread of virus from blood to other tissues causes the recruitment of inflammatory monocytes and natural killer (NK) cells (Figure 2), in association with the development of several clinical symptoms, including headache, back and muscle aches, nausea and vomiting, rash, arthralgia and myalgia [6,11,14,15]. However, despite certain clinical similarities, there are also differences in the most severe disease manifestations such as the prominent and prolonged arthralgia with CIKV infection, haemorrhagic fever with DENV infection and microcephaly and other neurological manifestations with ZIKV infection [1,6,16].

It is however important to note that infection by arboviruses induces a long-term protection, as recently demonstrated for CHIKV [17]. DENV is an excellent example of the opposite situation, with the presence of four different serotypes (DENV-1 to DENV-4). Infection with any of the four dengue serotypes induces protective immunity to the priming serotype but does not confer long-term protection against infection by other serotypes. Furthermore, heterotypic secondary DENV infection, with a DENV type distinct from the primary infecting type, is the greatest risk factor for Dengue haemorrhagic fever/dengue shock syndrome (DHF/DSS) [18]. It has been hypothesized that the worsening of disease following a secondary infection is due to antibody-dependent enhancement (ADE) (See below). Recently, Katzelnick et al. [19] demonstrated that the level of pre-existing anti-DENV antibodies is directly associated with the severity of secondary DENV disease in humans.

### 1.3. NK Cells: A Critical First-Line Defence of the Immune System against Viral Infections

NK cells constitute the proverbial first-line defence against a variety of viral infections because of their unique centred position in the immune system, NK cells are considered innate short-lived effectors with a turnover time of approximately 2 weeks, compared to months or years for some T cell subsets [20,21,22]. Upon activation and recruitment to the site of infection, the canonical function of NK cells is cytolysis. Consistent with the critical nature of NK cell-mediated killing, impaired cytolysis is the primary diagnostic criterion in patients with functional NK cell deficiencies, associated with herpes virus susceptibility [23,24]. NK cells also play both antiviral and regulatory roles via the release of soluble factors and operate via a balance of inhibitory and activating signals that enable them to detect and lyse virus-infected target cells while sparing normal cells [25,26].

Importantly, during acute infection the quantity of NK cells can significantly decrease in the peripheral blood, suggesting that NK cells are recruited from the periphery to sites of infection, in order to produce cytokines and become cytotoxic. The role of resident and circulating NK cells in the target tissues remains as yet elusive; however, the recent identification of markers specific for human tissue-resident NK cells, such as CD69, CD103 (αE integrin) and CD49 (α1 integrin) [27] is crucial for further studies on NK cells during viral infection.

Under normal immune surveillance, NK cells express inhibitory receptors, including killer Ig-like receptors (KIR-L), ILT-2 and the CD94:NKG2A heterodimer that recognize primarily classical and non-classical major histocompatibility complex (MHC) class I molecules. This “missing-self” recognition counterbalances cytotoxic CD8^+^ T-cell recognition restricted by pathogen-induced loss of MHC class I molecules. NK cells become functional when stimulatory signals, induced following the triggering of activating receptors, outweigh MHC class I-mediated inhibition. Several of these activating receptors have been characterized, including activating KIRs (KIR-S), which are highly homologous to their inhibitory counterparts in the extracellular domain, characterized by a short cytoplasmic tail lacking ITIMs, that interacts with DAP-12, a signalling polypeptide that induces NK cell activation. A growing number of studies point to a significant association between the presence of activating KIRs and the clinical outcome of some human diseases, including viral infections [28]. Interestingly, it was recently reported by Naiyer et al. [29] that HLA-C*0102 presents a highly conserved peptide derived from the helicase motif 1b region of related flaviviruses, including DENGV, ZIKV, YFV and Japanese encephalitis viruses, to KIR2DS2. The other activating NK receptors are: NKG2C, recently defined as the key receptor of the adaptive NK cells expended after viral infections, NKG2D and the natural cytotoxicity receptors (NCRs) NKp30, NKp44 and NKp46, in addition to specific co-receptors [26,30]. Their ligands are diverse and their expression mainly upregulated by cellular stress, which allows NK cells to specifically eliminate harmful or unhealthy host cells. The multitude of receptors that are expressed by defined NK cell subsets and that binds to viral ligands illustrates a specific mode of recognition.

The relevance of this recognition is highlighted by mechanisms evolved by viruses to evade the host immune system and diminish expression of stress ligands. This array of receptors also generates a vast diversity in the NK repertoire, arguing that single-cell diversity enhances their ability to fulfil this role. The diversity of the T- or B-cell repertoire is captured in a single antigen receptor, whereas NK cells, by contrast, are diverse at a single-cell level. Consequently, based on the expression of 28 cell surface receptors, the NK cell repertoire is composed of up to 3 × 10^4^ possible subpopulations [31].

The importance of the antiviral response mediated by NK cells in humans has essentially been indicated by indirect evidence, including the following:
(1)Most of the primary deficiencies in NK cell frequency or function are associated with an unusual susceptibility to herpes viruses [23,24].(2)KIR and HLA genotypes are associated with susceptibility or resistance to a large panel of viral infections, including arboviruses [32,33,34].(3)NK cells can control a viral infection in the absence of a T cell response. This was shown in a 3-month-old girl with a T^−^B^+^NK^+^ severe combined immunodeficiency phenotype who recovered from cytomegalovirus (CMV) infection without antiviral therapy. In this patient, the high number of NK cells (>2 × 10^10^/L) present at the peak of viremia, as well as the correlation between viral load and the number of NK cells during follow-up, were suggestive of a causal relation [35].(4)Adaptive NKG2C^+^ NK cells can be expanded in response to different viral infections in the context of human CMV seropositivity [36,37,38]. These hyper-reactive and long-lived NK cells have been described after certain viral infections and are able to mount stronger protective responses upon re-encounter of the same pathogen, at least in mice [39,40,41,42], which could redefine the notion of immunological memory for NK cells. These adaptive NK cells will be discussed in further detail later, in the context of arbovirus infection.

Altogether, these examples illustrate that NK cells can integrate a diversity of cues that contribute to their role during infection and in particular by arboviruses.

## 2. NK Cell Responses following Arbovirus Infection

### 2.1. Alphaviviruses

Alphaviruses are members of the *Togaviridae* family of single-stranded, positive-sense RNA viruses that make up a major group of medically important arboviruses. Alphaviruses impact human health around the world, often in areas with heavy disease burdens from other arboviruses that cause infections that present similar early clinical symptoms. Countries in the Americas and the Caribbean are experiencing a waning epidemic encompassing over 2 million suspected infections from the arrival of CHIKV in 2013, an alphavirus associated with chronic and debilitating polyarthralgias (Pan American Health Organization 2017. http://www.paho.org/hq/index.php?option=com_topics&view=readall&cid=5927&Itemid=40931&lang=en). In 2008, CHIKV was listed as a US National Institute of Allergy and Infectious Diseases category C priority pathogen. Persistent joint pain is a common symptom also caused by other related pathogen alphaviruses, such as RRV, O’nyong-nyong virus and Mayaro virus, which are poised to become the next emerging pathogen [11,43,44]. Intriguingly, outbreaks were mostly reported to be unpredictable, with an interval of several decades between major events; the precise factors of alphavirus emergence are unclear but it can hypothesized that, in addition to ecologic and virus-linked factors, the immune status of the affected populations plays a key role both in the intensity and periodicity of recurrence.

To date, the role of NK cells was mostly examined in detail for CHIKV infection. Although we and others have shown that NK cells mount an early and efficient protective response following CHIKV infection, their contribution to viral-induced pathology has been reported as well [45,46,47,48]. A few days after the onset of infection, NK cells display an activated profile [45,49] with increased expression of late-differentiation makers such as CD57, ILT-2 and CD8a. By contrast, expression of immature receptors such as NKG2A and CD161 is decreased together with the activating NKp30 and NKp46 receptors [45,46], as has been previously observed in other viral infections [20,50]. More intriguingly, CHIKV infection skews the NK cell receptor repertoire toward expression of the activating CD94/NKG2C receptor, which is usually very poorly or not at all expressed on NK cells of healthy donors. Initially, Guma et al. [51] described a skewing of the NK cell repertoire toward adaptive NK cells that express the activating CD94/NKG2C receptor in HCMV seropositive individuals. Similar observations have been made in the context of CHIKV [45], Hantavirus [52] and viral hepatitis B and C [53,54]. Until now, the presence and expansion of adaptive NK cells have almost exclusively been associated with prior exposure to HCMV or its reactivation [55]. This expansion seems partially driven by the increase in expression of HLA-E, the ligand of NKG2C, together with responsiveness of the cells to IL-12 and/or IL-15 [56,57]. In addition, several recent studies have demonstrated that sequence variation in peptides presented by HLA-E could affect activation and antibody-driven effector function of adaptive NK cells [20,58]. Although key questions regarding the regulation and function of adaptive NK cells remain, their expansion is clearly associated with the development of memory NK cells, at least in certain experimental animal models [59,60].

It is clear that NK cells in CHIKV infection are strongly cytotoxic without producing IFN-γ, thereby suggesting a prominent role of NK cells in the control and destruction of virus-infected target cells [45,47]. Paradoxically perhaps, the lack of IFN-γ in the microenvironment could alter or delay initiation of the adaptive immune response and be associated with a delay in the resolution of persistent symptoms during the chronic phase of infection [45]. Thus, in an experimental mouse model of CHIKV infection, extension of oedema was correlated with high NK activity, whereas the depletion of NK cells significantly reduced acute joint pathology, thus corroborating the deleterious role of NK cells in driving virus-induced pathology [61]. These data are consistent with those obtained in degenerative joint disease associated with osteoarthritis in which synovial tissue-infiltrating NK cells displayed impaired IFN-γ production, suggesting that NK cells contribute to the evolution of disease [62,63].

Overall, NK cells appear to be beneficial in the acute phase of CHIKV infection. Nevertheless, NK cells could contribute to immunopathologic mechanisms by infiltrating synovial tissues and maintaining an inflammatory environment that could contribute to the development of chronic articular inflammation in CHIKV-infected patients.

### 2.2. Flaviviruses

Among the most pathogenic flavivirus*es* in humans, the archetype, YFV, causes jaundice in those people who are severely affected. An effective and affordable vaccine providing long-lasting immunity has been widely available for more than 50 years [64]. This 17D vaccine is one of the most effective vaccines developed to date; 99% of recipients are protected for more than 10 years after a single vaccination (http://www.who.int/mediacentre/factsheets/fs100/en/). However, despite the success of this vaccine, YFV continues to circulate in the tropical jungles of Africa and South America and re-emerges regularly to infect naïve persons. WNV is even more widely dispersed because it infects a range of mosquito, mammalian, avian and reptilian species. This virus has attracted attention as a major pathogen following an outbreak of fever and encephalitis in New York, USA in 1999, where deaths in horses, birds and humans were reported. Within 5 years, the virus had dispersed throughout the Americas [65]. DENV fever remains an important disease caused by four closely related viruses, DENV 1–4, for which no DENV-specific therapies are available. The sole approved DENV vaccine elicits protection in people with prior DENV exposure but not in naïve individuals in which it has been associated with more cases of DENV haemorrhagic fever upon subsequent infection [66]. Another important question concerns ZIKV, which was confined as a low-pathogenic arbovirus for a long period of time but was recently declared as a public health emergency by the WHO. The reason for this dramatic change was its recent spread worldwide and its link to Guillain-Barré syndrome in adults and multiple neurodevelopmental defects, including microcephaly in infants born of mothers infected during the first trimester of pregnancy [16,66].

As described for other human pathogenic arboviruses, recovery from flavivirus infections depends on the host’s ability to mount effective innate antiviral responses that can eliminate or at least control, the virus. Following natural infection by an infected mosquito, viral particles are introduced into the subcutaneous space, where resident DCs are one of the first cell types to be infected by flaviviruses such as DENV or ZIKV. The DCs are specialized in the production of a robust level of type I IFN. Several lines of evidence have indicated that the type I IFN system is the central mediator of protection. First, studies with DENV-infected patient samples have shown that, during the early febrile period, high levels of type I IFN is observed in the serum of DENV-infected patients [67,68]. In addition to direct antiviral effects, type I IFN also mediates a variety of immunoregulatory effects, including regulation of NK development and maturation and activation of NK cells [69]. A subsequent local inflammatory response rapidly recruits leukocytes, such as type I-activated NK cells, to the site of infection [70], which is mediated by a modulation of the expression of specific homing markers in these cells. In particular a decreased expression of CXCR3 is associated with a concomitant increase of CCR10 in DENV^+^ patients, as compared to healthy donors; this homing of NK cells appears to be directed to the bone marrow, skin and possibly the lymph nodes [71]. The tissue-specific localizations are consistent with the generally accepted finding of DENV infection that often results in “petechiae” following resolution of infection A protective role of CXCR3-expressing NK cells appears to be important in DENV, is supported by the high mortality rates and viral load in the brains of CXCR3-deficient mice infected by DENV [72].

NK cells have been studied, to various extent, in several flavivirus infections, including DENV, WNV and ZIKV. Generally, in these infections, an early and robust activation of NK cells has been described, suggesting a key role for the latter cells. In DENV-infected patients, early activation of NK cells was also reported, clearly associated with mild dengue disease [15,73] and was also observed in children with DENV haemorrhagic fever [58]. In addition, an extensive phenotypic study has revealed a profound modulation of the NK cell repertoire, including downregulation of the activating receptors NKp30, NKp44, NKp46 and NKG2D in patients infected by DENV [73,74] or WNV [75]. A transient increase in the frequency of adaptive NKG2C^+^ NK cells was also found in DENV-infected patients, as previously described for CHIKV, whereas the most prominent response in WNV infection was observed among CD56^dim^CD16^−^ NK cells, an intermediate state of differentiation [76]. In a kinetics study, it was observed that the activation state of NK cells in DENV infection was short-lived [46], suggesting that there is no lasting imprint on the NK cell repertoire, as previously observed in CHIKV infection [45]. It is important to note that following vaccination with 17D, a live attenuated YFV, NK cell activation peaks at the same time as viral load, 6 days post-vaccination and correlates directly with a rise in plasma type I and type III IFNs. Thereafter, viral load and NK cell responses decline rapidly returning to baseline by day 10 and 15 post-vaccination, respectively [77,78].

In parallel, genetic evidence supports the role of KIRs in flavivirus infection. In studies performed in Gabon, Brazil and Western India, a difference in the frequencies of KIR genes was found in patients infected with DENV, as compared with healthy controls [33,79,80]. Mechanistically, KIR molecules interact with DENV peptides presented on HLA molecules. As observed in many viral infections [81], an interaction was demonstrated between KIR3DL1 and a conserved NS1 peptide of DENV presented by HLA-B57 [82]. By contrast, a peptide on the NS3 protein that is conserved in most flaviviruses, including DENV, stabilized HLA-C0102 and interacted with KIR2DS2 resulting in NK cell activation [29].

As in several other human viral infections (e.g., CMV, hantavirus, CHIKV and HIV-1), we observed an early accumulation of NKG2C^+^ NK cells in association with CD57 following acute infection by DENV-2 in Gabonese patients, in comparison to healthy donors. However, this increase in the frequency of CD57^+^NKG2C^+^ NK cells was transient and a return to normal levels occurred rapidly following the acute phase of infection [46]. Of note, in other flavivirus infections, such as WNV and tick-borne encephalitis (TBEV), another flavivirus transmitted to humans by ticks, specific expansion of NKG2C-expressing NK cells was never observed, even in CMV seropositive donors [75,83]. It remains possible that NKG2C^+^ NK cells expanded only locally, such as in the central nervous system in a TBEV infection. Indeed, this has been previously observed in patients with celiac disease, in which intraepithelial T cells in the gut lumen expressed high levels of NKG2C, whereas the corresponding T cells in the periphery remained unaffected [84].

The results from a principal component analysis based on the expression of different NK cell markers revealed that DENV-2 infection is mostly associated with the modulation of inhibitory KIRs and NKp44, a unique NK receptor only expressed on activated NK cells [85,86]. Consistently, a direct interaction of NKp44 and protein E of DENV and WNV has been observed [87]. These data suggest that this effect might be common for flavivirus infection regardless of the origin of the virus and E protein. Thus, it is likely that a complex balance exists between flavivirus-induced triggering of NK cells and subversion of NK killing as a result of the upregulation of the expression of MHC class I molecules on infected cells [15,88]. Increase of MHC class I expression is a hallmark of flavivirus activity, leading to the inhibition of NK cell function. In WNV infections, this modulation is associated with upregulation of NKG2D ligands in infected cultures, suggesting that these stress molecules could play a role in the recognition of infected cells [75]. Increase of MHC class I expression in ZIKV-infected cells is mediated via the RIGI-IRF3 pathway which can be inhibited by blocking IFN-β [89]. Other flaviviruses such as WNV are associated with activation of NF-κB, as shown after infection of IFNAR^(−/−)^ cells by WNV [90]. Furthermore, it was reported that HLA-E is upregulated in ZIKV but not in DENV-infected NK cells, suggesting a differential impact on NKG2A/2C responses [91]. At least in ZIKV, little or no changes in the expression of other ligands, such as AICL for NKp80; B7H6 for NKp30; CD48 for 2B4; and MHC class I chain-related proteins A and B (MICA and -B molecules for NKG2D) was shown on infected targets [89]. Because ligands of other NK receptors can be induced by “stress”, further studies are needed to characterize each NK receptor ligand that is induced during flavivirus infection.

In accordance with the upregulation of MHC class I expression, an increasing number of studies have also reported that flavivirus-infected target cells are unaffected over time, indicating that they are protected from NK cell cytotoxicity. This was shown by us in DENV-2-infected patients [46] and by Costa et al. in an optimized humanized mouse model, in which IFN-γ production by human NK cells was found to be important in controlling DENV replication in the absence of cytotoxicity [70]. Upregulation of IFN-γ production by NK cells was also observed in WNV infection both before and after NK cell expansion [92]. Similarly, ZIKV-infected cells were unaffected over time, indicating that they were protected from NK cell killing, whereas a strong, time-dependent secretion of IFN-γ from NK cells incubated with ZIKV-infected cells was shown [89]. In another study in Uganda, pre-existing IFN-γ-producing NK cells in an activated immune microenvironment were associated with lower viral loads following vaccination against YFV [78]. Nevertheless, it is important to note that an increased number of NK cells was reported in liver biopsies of patients who died of YFV [93].

There are two mechanisms by which a viral infection can be controlled: limiting the rate of production of new virus particles (by blocking viral entry or preventing the infected cell from releasing virus) or increasing the clearance of infected cells or virus. NK cells can play a role in the clearance of infected cells through antibody-dependent cell cytotoxicity (ADCC). In 1984, Kurane et al. [94] reported that NK cell-mediated lysis of Raji cells particles infected with the DV2 serotype was significantly higher when these cells had been preincubated with DV-immune serum. ADCC activity in plasma from school children in Thailand prior to a secondary infection with DENV appeared to correlate with neutralizing Ab titres and immune protection against secondary DENV-3 infection, although such a correlation was not observed with DENV-2 infection [95]. Recently, it was reported that the magnitude of ADCC, detected in non-neutralizing patient sera in DENV, is correlated with the expression of CD16 [96]. Polymorphism of CD16a in positions 48 and 158 has been reported to impact human IgG1 binding and ADCC activity and thus need to be examined in patients infected by DENV. Similarly, NK cells control WNV infection by detecting and lysing NS1-expressing-infected cells through ADCC [97].

In DENV, ADCC may be also important during secondary infections by another serotype, when serotype-specific neutralizing antibodies to DENV are present; the four DENV serotypes share 70% sequence homology and thus cross-reactivity is common, known to be an underlying mechanism of elevated immune-pathology associated with severe forms of DENV [98,99]. This can cause antibody-dependent enhancement (ADE), in which weakly neutralizing antibodies from the first infection bind to the second serotype and enhance infection of myeloid cells, such as monocytes and macrophages, expressing FcγR, like CD16 expressed by NK cells [100]. Recently, Sun et al. have reported the importance of NK cell-mediated ADCC to act as the first line of defence against ADE by showing that NK cell activation occurred concomitantly with the detection of ADE, with elevated expression of CD69 and NKp44 on NK cells [101]. In this study, a minimal activation of T cells was observed and it was furthermore shown that the magnitude of ADE in monocytes was up to tenfold increased when NK cells were depleted.

## 3. Conclusions

In this review, we sought to illustrate the role of NK cells in the control of arboviral infections. There are however significant limitations in our current understanding of the mechanisms by which human NK cells recognize the different arboviruses and mediate their antiviral function. Additional studies are needed to further elucidate the molecular mechanisms by which NK cells recognize infected target cells and the nature of the receptor-ligand families that are involved in this process.

## Figures and Tables

**Figure 1 viruses-11-00131-f001:**
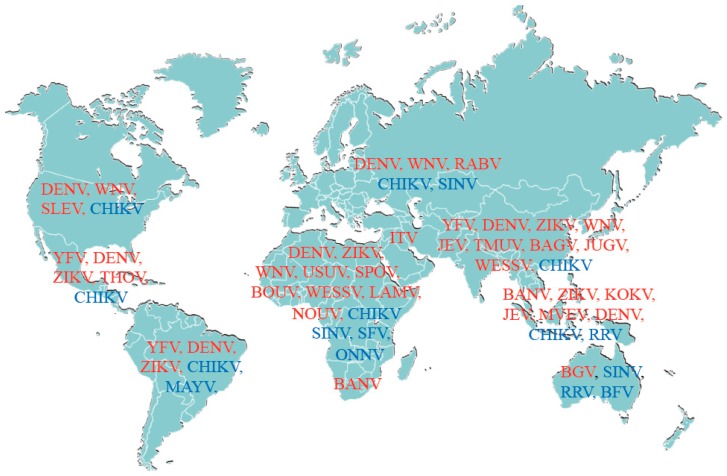
Non-exhaustive alphabetic list of flaviviruses (in red) and alphaviruses (in blue) and their geographical localization. Flaviviruses: Bagaza virus (BAGV), Bamaga virus (BGV), Banzi virus (BANV), Bouboui virus (BOUV), Dengue virus (DENV), Israel Turkey meningoencephalomyelitis (ITV), Japanese encephalitis virus (JEV), Jugra virus (JUGV), Kokobera virus (KOKV), Lamni virus (LAMV), Murray Valley encephalitis virus (MVEV), Nouanamé virus (NOUV), Rabensburg virus (RABV), Saint Louis encephalitis virus (SLEV), Spondweni virus (SPOV), Tembusu virus (TMUV), T’Ho virus (THOV), Usutu virus (USUV), Wesselsbron virus (WESSV), West Nile virus (WNV), yellow fever virus (YFV) and Zika virus (ZIKV). Alphaviruses: Barmah forest virus (BFV), Chikungunya virus (CHIKV), Mayaro virus (MAYV), O’nyong-nyong virus (ONNV), Ross River virus (RRV), Semliki forest virus (SFV) and Sindbis virus (SINV).

**Figure 2 viruses-11-00131-f002:**
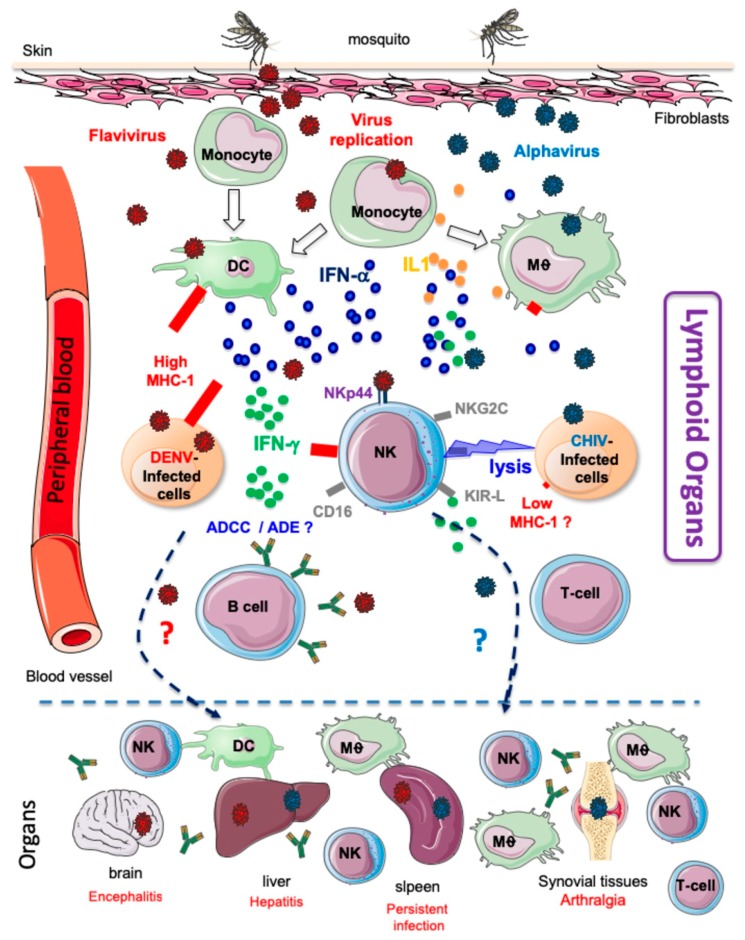
Virus dissemination, immune activation and clinical manifestations in patients infected by alphavirus or flavivirus. These viruses are transmitted through the bite of a female mosquito. The virus infects susceptible cells of the dermis, such as endothelial cells, fibroblasts and macrophages. Locally produced viral particles are then transported through the circulation to secondary lymphoid organs. This acute phase of infection is associated with the upregulation of the production of proinflammatory cytokines, such as IL-1β and IFN-α, that induce innate immune responses, including those exerted by NK cells. Through the circulation alphavirus or flavivirus disseminated also to different organs, including the brain, spleen, liver, joints and muscles. Peripheral and tissue-resident NK cells can directly fight infected cells by triggering cytotoxicity (alphaviruses) or massive production of IFN-γ (flaviviruses), which contributes to the control of infection and the generation of adaptive T cell immunity, as well as the production of protective antibodies and the induction of antibody-dependent cellular cytotoxicity. For other flaviviruses, such as DENV, antibody complexes are also associated with the triggering of antibody-dependent enhancement which has been linked to the development of more severe forms of disease.

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
