# Peer review of "Control of Acute Arboviral Infection by Natural Killer Cells"

_viruses, 2019, doi:10.3390/v11020131_

Round 1
Reviewer 1 Report
This short review provides timely coverage of the role of the innate immune compartment in protection or pathogenesis to Arbovirus infections. From the host perspective, particular emphasis has been placed on NK cells but also on the role of impacts of accessory cells relevant to the NK cell response. From an Arbovirus perspective, the article takes has its principal focus on alphavirus - CHICKV and flavivirus - DENV, ZIKV infections.
1. A more structured introduction to inhibitory and activating NK cell receptors, particularly those subsequently covered in the context of Arbovirus/ Flavivirus infection would be helpful. For example particular introducing short form activating KIR should be introduced alongside inhibitory KIR, including what the nomenclature implies in terms of extracellular domains and signalling motifs. This is important in the context of studies quoted subsequently in lines (273-281)
2. It is important to emphasise that NKG2C expansions and the generation of ‘adaptive’ (FceRg-/ PLZF- NK cells within these are most strongly and primarily associated with HCMV infections. This is important because:
Unlike the most other infections HCMV stabilises a direct ligand- receptor interaction for NK cell activation, which has not been clearly demonstrated for the other infections.
Mechanistically it may be most relevant that indirect effects of subsequent arbovirus infections imparted via inflammatory cytokines which alone or in conjunction with direct interaction of viral peptides and NK cell activating receptors, may boost the expansion and function of NKG2C+ NK cells in HCMV infected individuals. This is demonstrated, for example in Hantavirus infection where IL-15 responsiveness may play a role (Quote the following papers in the relevant sectiosn where the example of Hantavirus mediated expansions are give: Bjorkstrom et al J Exp Med. 2011 Jan 17;208(1):13-21. doi: 10.1084/jem.20100762. and Braun et al PLoS Pathog. 2014 Nov 20;10(11):e1004521. doi: 10.1371/journal.ppat.1004521).
3. One assumes that the main interactions between host cells during viral infection are initiated in the skin and progress in lymphoid tissue or in the case of pathogenesis, in relevant target organs and not in the blood. The authors need to distinguish more clearly where the distinct cellular interactions may be taking place –compartmentalising more clearly to the skin in the initiation phase and then within the relevant lymphoid and non-lymphoid tissues thereafter. Figure 2 gives the impression that most of the cellular interactions occur in the blood vessels and this should be adjusted to the localisation of the innate responses more precisely.
4. Some more precise detail about what is known about the normal phenotype of NK cells in SLT and infiltrates in diseased tissues should be provided. Better information linking NK cell phenotype to disease associated migratory patterns would also be helpful in this context. In Lines 282-286 link transient changes during acute DENV-2 infection to likely depletion of less differentiated NK cells form the circulation.
5. Lines 324-355. Some more detail on NK cell ADCC should be provided here, including linking CD16 as the major NK Fc receptor and how the IgG isotype binding profile links to what is known about those promoting ADE. How ADCC function links to the NK cell differentiation phenotype and expansions of particular NK cell subsets mentioned earlier should also be covered.
6. There are some minor issues of word usage which should be reviewed and corrected, for example in line what does the word 'decorticated' mean?
Author Response
Reviewer 1
This short review provides timely coverage of the role of the innate immune compartment in protection or pathogenesis to Arbovirus infections. From the host perspective, particular emphasis has been placed on NK cells but also on the role of impacts of accessory cells relevant to the NK cell response. From an Arbovirus perspective, the article takes has its principal focus on alphavirus - CHICKV and flavivirus - DENV, ZIKV infections.
We thank very much the Reviewer 1 for this positive and helpful review. We feel that the comments that were made have helped us substantially to improve the manuscript. In response to the concerns:
1. A more structured introduction to inhibitory and activating NK cell receptors, particularly those subsequently covered in the context of Arbovirus/ Flavivirus infection would be helpful. For example particular introducing short form activating KIR should be introduced alongside inhibitory KIR, including what the nomenclature implies in terms of extracellular domains and signalling motifs. This is important in the context of studies quoted subsequently in lines (273-281)
Author response: As request by Reviewer 1, activating KIRs (KIR-S) have been added in the introduction on NK cells in line with arbovirus infections (Line 139), as well as specific references
2. It is important to emphasise that NKG2C expansions and the generation of ‘adaptive’ (FceRg-/ PLZF- NK cells within these are most strongly and primarily associated with HCMV infections. This is important because:
Unlike the most other infections HCMV stabilises a direct ligand- receptor interaction for NK cell activation, which has not been clearly demonstrated for the other infections.
Mechanistically it may be most relevant that indirect effects of subsequent arbovirus infections imparted via inflammatory cytokines which alone or in conjunction with direct interaction of viral peptides and NK cell activating receptors, may boost the expansion and function of NKG2C+ NK cells in HCMV infected individuals. This is demonstrated, for example in Hantavirus infection where IL-15 responsiveness may play a role (Quote the following papers in the relevant sectiosn where the example of Hantavirus mediated expansions are give: Bjorkstrom et al J Exp Med. 2011 Jan 17;208(1):13-21. doi: 10.1084/jem.20100762. and Braun et al PLoS Pathog. 2014 Nov 20;10(11):e1004521. doi: 10.1371/journal.ppat.1004521).
Author response: Thank you for this comment. In agreement with Reviewer 1 more details have been added concerning the mechanism of NKG2C expansion in line 211 as well as specific references
3. One assumes that the main interactions between host cells during viral infection are initiated in the skin and progress in lymphoid tissue or in the case of pathogenesis, in relevant target organs and not in the blood. The authors need to distinguish more clearly where the distinct cellular interactions may be taking place –compartmentalising more clearly to the skin in the initiation phase and then within the relevant lymphoid and non-lymphoid tissues thereafter. Figure 2 gives the impression that most of the cellular interactions occur in the blood vessels and this should be adjusted to the localisation of the innate responses more precisely.
Author response: In agreement with the Reviewer 1, the text has been modified (Line 75) as well as the Figure 2 and its legend
4. Some more precise detail about what is known about the normal phenotype of NK cells in SLT and infiltrates in diseased tissues should be provided. Better information linking NK cell phenotype to disease associated migratory patterns would also be helpful in this context. In Lines 282-286 link transient changes during acute DENV-2 infection to likely depletion of less differentiated NK cells form the circulation.
Author response: Thank you for this important question concerning the possible migration of the NK cells in the tissues (and the putative role of resident NK cells) in relationship with transient changes of NK cells in the circulation during acute viral infection. A specific comment has been added in Line 125, as well as specific new references concerning the identification of resident tissue-specific NK cells.
5. Lines 324-355. Some more detail on NK cell ADCC should be provided here, including linking CD16 as the major NK Fc receptor and how the IgG isotype binding profile links to what is known about those promoting ADE. How ADCC function links to the NK cell differentiation phenotype and expansions of particular NK cell subsets mentioned earlier should also be covered.
Author response: More detail on ADCC and ADE have been added in Lines 361 and 380, respectively, as well as new references.
6. There are some minor issues of word usage which should be reviewed and corrected, for example in line what does the word 'decorticated' mean?
Author response: The revised version of the manuscript has been completely edited to correct some mistakes
Reviewer 2 Report
In Manuscript ID: viruses-417387 Maucourant and colleagues give an overview on the role of natural killer cells in the control of acute arboviral infections. The manuscript is well organized and easy to read. It gives a nice update on different aspects of NK cell biology in the context of arboviral infections.
Some minor comments:
- line 66-67 'increases in globalization have resulted in increases in the spread of diseases to population lacking native immunity, resulting in major economic consequences'. It is not clear that native immunity is important in the defense against arbovirla infections, and if so what part of the immune system is meant. Please explain.
- line 142: the nomenclature of "adaptive NKG2C+ NK cells"is confusing since NK cells are not adaptive cells but cells that have similar properties as some cellular subsets of the adaptive immune system. Referring to this population of NK cells as memory NK cells, or long lived NK cells is more clear
- from line 323 onwards: the role of ADCC is mentioned, but very briefly compared to the rest of the manuscript. This section should be expanded, for example the information in line 333-335 could be specified.
- in line 342 the authors refer in their conclusion to the development of targeted interventions. This is however not addressed in the manuscript. What kind of interventions t? Any examples from other NK therapies that may be applied in during infections with arboviruses? Please explain.
Author Response
Reviewer 2
Comments and Suggestions for Authors
In Manuscript ID: viruses-417387 Maucourant and colleagues give an overview on the role of natural killer cells in the control of acute arboviral infections. The manuscript is well organized and easy to read. It gives a nice update on different aspects of NK cell biology in the context of arboviral infections.
We thank very much the Reviewer 2 for this very positive review and the different suggestions to improve the quality of our manuscript. In response to the minor concerns:
Some minor comments:
- line 66-67 'increases in globalization have resulted in increases in the spread of diseases to population lacking native immunity, resulting in major economic consequences'. It is not clear that native immunity is important in the defense against arbovirla infections, and if so what part of the immune system is meant. Please explain.
Author response: A specific comment has been added in Line 100 concerning the question of the protection and the heterotypic secondary infection.
- line 142: the nomenclature of "adaptive NKG2C+ NK cells"is confusing since NK cells are not adaptive cells but cells that have similar properties as some cellular subsets of the adaptive immune system. Referring to this population of NK cells as memory NK cells, or long lived NK cells is more clear
Author response: I am not totally agreed with this comment: The definition of an “adaptive NK cells” seems to be well accepted by the NK-cell community. In contrast, the possibility that memory NK cells could exist in all infections associated with adaptive NK cells, including arboviral infections, is not demonstrated yet.
- from line 323 onwards: the role of ADCC is mentioned, but very briefly compared to the rest of the manuscript. This section should be expanded, for example the information in line 333-335 could be specified.
Author response: Thank you for this comment; In agreement with Reviewer 2, the section dedicated to the ADCC has been expended in the revised version of the manuscript (Line 361).
- in line 342 the authors refer in their conclusion to the development of targeted interventions. This is however not addressed in the manuscript. What kind of interventions t? Any examples from other NK therapies that may be applied in during infections with arboviruses? Please explain.
Author response: In agreement with reviewer 2, the last sentence on an opening on therapies targeting NK cells has been deleted in the revised version of the manuscript.